# Fullerenes on a Nanodiamond Platform Demonstrate Antibacterial Activity with Low Cytotoxicity

**DOI:** 10.3390/pharmaceutics15071984

**Published:** 2023-07-19

**Authors:** Olga Bolshakova, Vasily Lebedev, Elena Mikhailova, Olga Zherebyateva, Liliya Aznabaeva, Vladimir Burdakov, Yuri Kulvelis, Natalia Yevlampieva, Andrey Mironov, Igor Miroshnichenko, Svetlana Sarantseva

**Affiliations:** 1Petersburg Nuclear Physics Institute Named by B.P. Konstantinov, NRC “Kurchatov Institute”, 188300 Gatchina, Russia; bolshakova_oi@pnpi.nrcki.ru (O.B.); lebedev_vt@pnpi.nrcki.ru (V.L.); burdakov_vs@pnpi.nrcki.ru (V.B.); kulvelis_yv@pnpi.nrcki.ru (Y.K.); 2Department of Microbiology, Virology, Immunology, Faculty of Preventive Medicine, Orenburg State Medical University (OrSMU), 460000 Orenburg, Russia; lelenaalekseevna@yandex.ru (E.M.); fenixmihail@yandex.ru (O.Z.); lkhus@yandex.ru (L.A.); 3Physical Faculty, St. Petersburg State University, 199034 St. Petersburg, Russia; n.yevlampieva@spbu.ru; 4G.N. Gabrichevsky Moscow Research Institute of Epidemiology and Microbiology, 125212 Moscow, Russia; beljamen@yandex.ru; 5Department of Normal Physiology, Faculty of General Medicine, Orenburg State Medical University (OrSMU), 460000 Orenburg, Russia; k_microbiology@orgma.ru

**Keywords:** fullerene, nanodiamond, polymer, cell, microorganism, antibacterial activity, biofilms, toxicity

## Abstract

Carbon nanoparticles with antimicrobial properties, such as fullerenes, can be distinguished among the promising means of combating pathogens characterized by resistance to commercial antibiotics. However, they have a number of limitations for their use in medicine. In particular, the insolubility of carbon nanoparticles in water leads to a low biocompatibility and especially strong aggregation when transferred to liquid media. To overcome the negative factors and enhance the action of fullerenes in an extended range of applications, for example, in antimicrobial photodynamic therapy, we created new water-soluble complexes containing, in addition to C60 fullerene, purified detonation nanodiamonds (AC960) and/or polyvinylpyrrolidone (PVP). The in vitro antibacterial activity and toxicity to human cells of the three-component complex C60+AC960+PVP were analyzed in comparison with binary C60+PVP and C60+AC960. All complexes showed a low toxicity to cultured human skin fibroblasts and ECV lines, as well as significant antimicrobial activity, which depend on the type of microorganisms exposed, the chemical composition of the complex, its dosage and exposure time. Complex C60+PVP+AC960 at a concentration of 175 µg/mL showed the most stable and pronounced inhibitory microbicidal/microbiostatic effect.

## 1. Introduction

The search for antimicrobial agents is becoming increasingly important due to the growing resistance of viruses and bacteria to the commercial drugs that are used at present. Fullerene is emerging as an important material for effective antimicrobial preparations [1], including the means of disinfection and sterilization of environments and surfaces contaminated with micro-organisms. The antimicrobial properties of fullerenes are due to the fact that they are membranotropic substances that easily penetrate into microorganisms, interacting with cellular structures. When excited by light, fullerenes exhibit the properties of photosensitizers that generate a reactive singlet oxygen (SO), which causes cell death, and this effect underlies photodynamic therapy [2,3,4,5,6]. It is noteworthy that the C60 fullerene possesses the highest quantum yield of singlet oxygen (96%) [7], and the adaptation of microorganisms to chemotherapy when using photodynamic exposure is practically impossible [1,8,9].

However, for use in biology and medicine, water-insoluble fullerenes must be functionalized [10]. At the same time, functionalization, including the addition of hydrophilic groups, molecules, oligomers, and polymers [11,12,13,14], must retain the properties and biological activity inherent in fullerenes. Fullerenes can be modified with amino groups for better binding to the surface of bacteria [15], carboxyl groups [16], and used to synthesize complex compounds with antimicrobial properties [17,18]. A combination of fullerene derivatives with antimicrobial agents greatly enhances the effectiveness of the latter [19]. Fullerenes modified with cationic groups showed a high photodynamic activity against a wide range of microorganisms. The hydrophobic carbon shells of modified fullerenes contributed to their penetration through cell membranes, and positive charges on the shells enhanced the binding of fullerenes by microbial cells [9]. For photodynamic antibiotic therapy, a hydrogel was obtained by the peptide-modulated self-assembly of fullerenes, preventing their aggregation by non-covalent binding to the peptide. It has been shown that peptide–fullerene hydrogels are more effective than the original fullerene in inhibiting antibiotic-resistant *Staphylococcus aureus* and promoting wound healing [20]. For the photoinactivation of microorganisms, photodynamic polymers with fullerenes have also been created [21].

An alternative way is the association of fullerenes with hydrophilic nanoparticles, such as nanodiamonds (NDs), which can serve as a platform for the transfer of fullerenes into aqueous media. It should be noted that ND particles themselves also exhibit antimicrobial activity [22,23,24], presumably due to the deformation and destruction of cell walls [25] that protect microorganisms from osmotic changes [26], as well as due to oxidative stress [22]. The antibacterial activity of nanodiamonds can be due to the negatively charged acid anhydride groups present on the surface [27,28]. Nanodiamonds are effective not only against planktonic cells, but also against biofilms [29]. For example, mannose-modified NDs have in particular shown a high potential in countering *E. coli*-biofilm formation [30]. ND surface modification methods [31] include a functionalization of diamonds by grafting various groups (H, OH, and COOH), which makes it possible to create stable aqueous diamond dispersions by controlling the sign and magnitude of their surface potential [32]. Such hydrophilic diamonds can serve as universal active platforms for the delivery of fullerenes and other hydrophobic molecules (drugs) to biological media. At the same time, NDs are radiation-resistant and chemically inert, which makes it possible to activate fullerenes not only by visible light or UV, but also by higher-energy quanta (X-rays) with a great penetrating power. It is important to create new photosensitizers (PS) based on fullerenes and diamonds, since diamonds are capable of X-ray luminescence. In addition, they can serve as converters of hard radiation into visible light for the excitation of optically active molecules attached to them, which is necessary for the implementation of X-ray photodynamic therapy (XRD) [33,34,35].

Especially, it is of interest to create C60 and ND complexes additionally stabilized with a medical polymer polyvinylpyrrolidone (PVP), as new antimicrobial agents with high photodynamic properties against antibiotic-resistant microorganisms. The PVP is a non-toxic, non-ionic polymer containing a hydrophilic pyrrolidone moiety and a hydrophobic carbon chain. It is able to serve as a model of natural macromolecules with amide groups (protein molecules), a carrier of biologically active substances, and a component of drugs due to low toxicity and biocompatibility [36,37]. PVP macromolecules in aqueous solutions have a high ability to form complexes with inorganic ions, organic molecules, and natural and synthetic polymers, which is of great interest for chemical, biochemical, and pharmaceutical technologies, especially in connection with the ability to combine PVP with carbon nanoparticles (fullerenes, nanotubes, graphenes, and diamonds) [38,39,40,41].

In this regard, the aim of our work is to study the antibacterial action and toxicity of the first prepared triple hydrophilic complex consisting of C60, ND, and PVP in comparison with the action of the binary complexes C60-PVP and C60-ND.

## 2. Materials and Methods

### 2.1. Preparation of the Samples of Carbon Nanoparticles

To create the complexes, we used purified detonation nanodiamonds (our notation AC960, produced by Hongwu International Group Ltd. (Guangzhou, China), www.hwnanomaterial.com (accessed on 12 July 2020), mark C960 with minor impurities). C60 fullerenes (99.9% wt. of main component with residual higher homologues) were synthesized from graphite in an electric arc discharge process with subsequent extraction from carbon soot and chromatographic purification (PNPI) [42].

To obtain the complexes of fullerenes with diamonds, ultrasonic stirring (US, 20 °C) was used to dissolve separately C60 powder in o-xylene and PVP (molecular weight of 9.5 × 103) in dimethyl sulfoxide (DMSO). The solutions were then mixed and sonicated to enhance the interaction and bind the components into primary complexes.

Under the same conditions, AC960 powders (quite spherical particles of ~10 nm in size; carbon is not less than 99% wt, impurities of Ni (<0.0005%), Mg (0.005%), Fe (0.005%), Cu (0.0007%), Na (<0.0005%), Ca (0.0042%)) were dispersed in DMSO by sonication. Then, the diamond colloid was mixed with the solution of C60+PVP complexes, and C60+PVP+AC960 system was prepared. The ternary colloid was dried to remove DMSO completely and further dissolved again in deionized water to produce the target product being an aqueous colloid of the C60+PVP+AC960 complex (Sample 1). Binary aqueous systems C60+PVP and C60+AC960 were prepared in DMSO in a similar way. These colloids were dried and then dissolved in water to obtain the reference samples C60+PVP (Sample 2) and C60+AC960 (Sample 3).

The prepared aqueous systems (Samples 1–3) had the total concentrations of 0.7, 0.5, and 0.4 mg/mL. To saturate the samples with fullerenes, the mass proportions for the components for the synthesis were initially preset:

Sample 1: C60 (11.1%)+PVP (44.4%)+AC960 (44.5%);

Sample 2: C60 (20%)+PVP (80%);

Sample 3: C60 (20%)+AC960 (80%).

### 2.2. Particle Size Analysis in Samples and the Stability Test of the Complex

Nanoparticle Tracking Analysis (NTA) is the method for analyzing particle size distribution in liquids, based on relating Brownian motion to the hydrodynamic diameter of the single particle. The size distribution and concentration were analyzed by NanoSight LM10 (Malvern Panalytical, Malvern, UK), which utilizes NTA, using a cuvette with a 405 nm laser (Nano-Sight, Malvern Panalytical, Malvern, UK). To perform better measurement, the suspension was diluted up to 100,000 times to obtain the optimum concentration. Measurements were performed in triplicates of 60 s video captures with a camera level 16, the lowest expectable particle size = 30 nm, and the detection threshold = 7 throughout the characterization. Video processing, size/concentration analysis, and statistics were conducted using the NanoSight NTA 2.3 software (Malvern Panalytical, Malvern, UK).

We performed the tests to assure the stability of complexes in the physiological conditions and measured the optical density of the aqueous solutions of complexes by spectrophotometer DU-8600RN(PC) (Drawell Scientific, Yuzhong District, Chongqing, China) at ambient temperature (23–24 °C) and then heated the solutions up to 37 and 50 °C, followed by cooling to 25 °C. For the ternary and binary complexes at the enhanced temperatures and final one, we evaluated a variation in optical density vs. light wavelength regarding the initial temperature. Along with this, we stored the samples 1–3 at ambient temperature (dark conditions) for three months and measured the optical density of these solutions; the final states were compared with the initial one.

### 2.3. Concentrations of Test Substances and Evaluation of Sterility

To assess the antibacterial activity of the samples, the following sample concentrations were used: sample 1: 350 µg/mL (C60—38.9 µg/mL, PVP—155.4 µg/mL, AC960—155.7 µg/mL) and 175 µg/mL (C60—19.4 μg/mL, PVP—77.7 μg/mL, AC960—77.9 μg/mL); sample 2: 250 μg/mL (C60—50 μg/mL, PVP—200 μg/mL) and 125 μg/mL (C60—25 μg/mL, PVP—100 μg/mL); and sample 3: 200 μg/mL (C60—40 μg/mL, AC960—160 μg/mL), and 100 μg/mL (C60—20 μg/mL, AC960—80 µg/mL). To analyze toxicity in cell culture, concentrations from 1 μg/mL to 300 μg/mL were selected depending on the experimental conditions. Since ultrasound was used to prepare the preparations, the solutions were not sterilized. To confirm the sterility of the colloids of the studied compounds, the microscopy of the samples and their inoculation on a rich dense nutrient medium (Columbia agar with 5% sheep blood, bioMerieux SA, St. Louis, MO, USA) were conducted. All colloids studied, when tested for sterility, did not show growth of any microorganisms on a nutrient medium. Facultative anaerobic microorganisms were chosen as objects for the experiment. Therefore, since the cultivation of these microorganisms does not require the creation of anaerobic conditions, it was decided to consider the suitable colloids according to microbiological criteria to implement the goals of the experiment.

### 2.4. Cell Cultures

Human skin fibroblasts (cell line FD2) and ECV cells (human umbilical vein endothelial cells) (Institute of Cytology of the Russian Academy of Sciences, Saint-Petersburg, Russia) were used as an in vitro model for toxicity analysis. Cells were cultivated in DMEM F12 with glutamine (Biolot, St. Petersburg, Russia), bovine serum 10% (Biolot, St. Petersburg, Russia), and fibroblast growth factor (for line FD2) at a concentration of 20 ng/mL (cat#PSG060-10. Lot#16F0519F2, Sci-Store, Skolkovo, Russia). Cells were incubated at 37 °C in 5% CO_2_. To analyze the morphology, ECV cells grown on coverslips were fixed with 96% ethanol, stained for 30 s with Mayer’s hematoxylin (BioVitrum, St. Petersburg, Russia), and analyzed using a Leica DM 2500 (Leica microsystems, Wetzlar, Germany) light microscope. The MTT assay and apoptosis assay were used to determine toxicity.

### 2.5. MTT Assay

Fibroblasts were seeded in 48-well plates and incubated until reaching 60–70% confluency. Then, the medium was changed to the other one containing the samples with the concentrations 1, 10, 50, and 100 μg/mL (200 μL per well) and the cells were cultured for another 24 or 48 h. Samples with the concentrations of 175–300 μg/mL were added for 45 or 90 min. Then, 20 μL of the MTT working solution (5 mg/mL) was added to each well (except for those marked as “blank”) and incubated under the same conditions for 2 h. The medium was then removed, the cells were gently washed with fresh medium, and 200 μL of dimethyl sulfoxide (DMSO) was added to all wells (including the “blank” ones) to dissolve the formazan granules. After the 30 min incubation at room temperature needed for a complete dissolution of the formazan crystals, the solution was transferred to a 96-well plate (100 μL per well).

ECV cells were plated in 96-well plates (10,000 cells per well), incubated for 24 h in a 5% CO2 atmosphere at 37 °C. Then, we added the samples. Analysis was conducted as described elsewhere [43].

The optical density (OD) of the contents of the wells was measured (Multiskan FC spectrophotometer, Thermo Scientific (Waltham, MA, USA), wavelength of 540 nm). The cell viability in the experimental samples was found as the percentage of the control by the formula (%) = [OD experiment − OD blank]/[OD (control) − OD blank] × 100%. OD blank data were used to take into account the contribution of the samples to the total OD magnitudes.

### 2.6. Apoptosis Assay

Cell death was analyzed by flow cytometry. The cells were seeded in 6-well plates and incubated as in the previous experiment. Then, the samples were added to obtain the concentration of 10 or 100 μg mL in the final mixtures for 48 h. Cells without added compounds were used as the experimental control. The treatment and analysis were conducted as described elsewhere [44].

### 2.7. Research Object of Antimicrobial Activity

Clinical isolates of staphylococci (*Staphylococcus aureus*), enterobacteria (*Escherichia coli*), fungi of the genus Candida, as well as bacilli (*Bacillus subtilis*) from the collection of the Department of Microbiology, Virology, Immunology of the OrSMU were used as the biological test objects. The assessment of the biochemical activity of microorganisms was conducted using the MIKRO-LA-TEST test systems (Lachema, Brno, Czech Republic). Standardized MALDI-TOFMS methods (VitekMS, bioMérieux, Craponne, France) were used for the species (genus-level) identification of the tested microorganism by the comparison of the protein spectrum of the studied strain to the collection of reference spectra of the known species contained in the database.

### 2.8. Cultivation of Microorganisms

*Escherichia* was cultivated on agar and Muller–Hinton broth (Himedia, Mumbai, India) for 18–20 h at 37 °C. Staphylococcus was cultivated using Nutrientagar and Nutrientbroth (Himedia, India) and the GRM medium (Obolensk, Moscow, Russia) for 18–20 h at 37 °C. Nutrientagar and Nutrientbroth (Himedia, India) were used for Bacillus cultivation, and the culture was grown for 18–24 h. The cultivation of fungi of the genus Candida was conducted on Sabouraud’s medium at 30–37 °C for 48 h.

### 2.9. Sample Preparation of Microbial Cultures

To study the effect of the tested compounds on the biological properties of microorganisms, microtubes were prepared, in which the necessary ingredients were added (a suspension of microorganisms and various concentrations of the studied compounds). At the stage of logarithmic growth, 100 mL of cultures of microorganisms in broth were mixed with the substance under study (the volume of the matrix solution depends on the required final concentration) and the broth was adjusted to a final sample volume of 200 mL. In all samples, the same series of liquid nutrient medium was used as at the stage of obtaining the cell sediments of the studied cultures. This was conducted to eliminate the stress associated with changing the chemical composition of the medium used. Samples in microtubes were incubated for 45 min and 90 min. The time of co-incubation was chosen according to the literature [45,46]. It was conducted according to the ability of the studied compounds to influence the degradation of the biofilm matrix. After co-incubation for a predetermined time, the tubes were centrifuged at 3000 rpm, followed by the supernatant removal, and the cell sediment was washed 3 times with saline NaCl to remove the trial compounds.

### 2.10. Evaluation of the Antibacterial Effect

The antibacterial effect was assessed by the indicators of the inoculation of microorganisms from the samples. For this purpose, the seeding of each sample was conducted with a microbiological loop using the method of depleting inoculation on a dense nutrient medium. The culture medium contained 5% sheep erythrocytes (Columbia agar with 5% sheep blood, bioMerieux SA). The cultures were incubated for 18–24 h at 37 °C. The result was taken into account by counting the number of grown colonies on the experimental and control plates [47].

### 2.11. Biofilm Analysis

Into 96-well sterile flat-bottomed plates (Medpolimer, St. Petersburg, Russia), 100 µL of a daily broth culture suspension of microorganisms was added simultaneously. The microorganisms *Staphylococcus aureus* and *Escherichia coli* were used at a concentration of 0.5 × 10^6^ cfu/mL. Additionally, 100 µL of the liquid nutrient medium was added as a control. Then, the complexes were added in the same volume at the previously described concentrations. Samples were incubated at 37 °C statically for 24 or 72 h. After the removal of planktonic cells, the wells were washed three times with PBS (pH 7.4) (200 μL each). The plates were dried in air for 30 min. Then, the formed biofilms were stained for 45 min at room temperature (22 °C–25 °C) with 1% crystal violet solution according to the method [48]. The free dye was washed three times with PBS (pH 7.4). The biofilm-fixed dye was then extracted with 96% ethanol (200 μL per well). The extract was transferred from the wells into clean 96-well sterile flat-bottomed plates (Medpolimer, Russia). The intensity of staining corresponding to the degree of biofilm formation was measured spectrophotometrically on a plate densitometer Multiskan Ascent (Thermo Electron corporation, Vantaa, Finland) at a wavelength of 540 nm.

### 2.12. Statistical Analysis

Statistical analysis was performed using KyPlot 6.0 software. A Dunnett’s test was used for a comparison of two samples (toxicity and antimicrobial activity) and the Tukey–Kramer test was used for multiple comparisons (to analyze biofilm formation). Data are presented as mean ± SET (standard deviation) or mean ± SEM (standard error of the mean). Differences at *p* < 0.05 were considered statistically significant.

## 3. Results

### 3.1. Characteristics of the Complexes

To maximize the use of the antibacterial activity of fullerenes, in the preparation of the samples, the option of obtaining complexes was chosen. This is due to the fact that the fullerenes in them do not form chemical bonds with components and retain their original physical and chemical properties to the greatest extent. In complexes, fullerenes are not aggregated and can be released from them, penetrate into the membranes of microorganisms, disrupt their metabolism, and inhibit growth.

During the preparation of the complexes, the abilities of the polymer and diamond to bind fullerene were significantly different. PVP is able to form a complex with C60 by electron transfer from the carbonyl group of the chain unit to fullerene. Earlier, this made it possible to obtain compositions by the solution or solid-phase methods with the achieved fullerene fraction up to 1% and 2.0–2.3% wt., respectively [14].

Complexes of this kind make it possible to achieve the finest dispersion of fullerenes in an aqueous medium according to the data of hydrodynamic methods and neutron scattering [14,49,50]. Due to the specific nature of detonation synthesis and subsequent chemical purification, the used diamonds had carbonyl, carboxyl, hydroxyl, and other groups on the surface [32]. The presence of groups created opportunities for bonding diamonds with fullerenes, which was also enhanced by the hydrophobic interactions of the components.

Thus, both the polymer and diamond components were capable of associating with fullerenes. This was confirmed (Figure 1A) in the measurements of the optical density of diluted aqueous colloids in the samples 1–3 in the wavelength range λ = 190–1100 nm (UV-VIS Spectrophotometer DU-8600RN(PC)). The spectrum for o-xylene fullerene with the characteristic maximum at the wavelength λmax~335 nm is also shown in the figure. The binding of fullerene with PVP led to a shift of the fullerene absorption band towards longer wavelengths (λmax~370 nm), indicating a change in the electronic structure of fullerene due to the interaction with polymer. In the same position on the curve for the ternary complex, there is a weak increment in optical density against the background of a growing contribution from diamonds at shorter wavelengths. This confirms a transfer of fullerenes into the aqueous solution by means of complexes. However, this spectral effect is almost imperceptible in the case of the C60+AC960 system without PVP when the association of fullerenes with diamonds is not revealed (Figure 1A).

More clearly, fullerenes binding to diamonds in comparison with the result when using PVP are shown in Figure 1B, where the curves for the modified optical densities D•λ are plotted for the C60+AC960 and C60+PVP+AC960 complexes. These data correspond to the absorption coefficient, multiplied by the wavelength.

In the spectrum for the ternary complex, a characteristic fullerene peak in the band of 320–420 nm is clearly visible while in a shifted position (λmax~370 nm) relative to the similar peak for C60 in o-xylene (Figure 1B). For the C60+AC960 complex in the same λ range, a small contribution of fullerene to the optical density is manifested against the background (Figure 1B). A detailed comparison of the peak amplitudes for complexes and fullerene in xylene, taking into account the concentrations of the dissolved substances, made it possible to estimate the actual fractions of fullerene (CF) in the dry binary and ternary complexes: CF = (2.2 ± 0.2) % wt., C60+PVP; CF = (0.6 ± 0.2) % wt., C60+AC960; and CF = (1.2 ± 0.2) % wt., C60+PVP+AC960.

The evaluated amounts of fullerenes in water-soluble complexes are by an order of magnitude lower than the concentrations of C60 taken initially to prepare the compositions. Finally, the ternary complex C60+PVP+AC960 has, on average, ~two fullerenes on a diamond particle surface. In the case of the direct association of fullerenes with diamonds in the C60+AC960 complex, only ~one C60 molecule was attached to the diamond particle. The number of fullerene molecules in the complex with PVP was ~one C60 molecule per ~ three polymer chains (~300 monomer units), providing fullerene shielding from aqueous environment.

Thus, ternary complexes allowed us to obtain the molecular dispersion of fullerenes in the aqueous medium due to the polymer shell around C60 molecules and their attachment to diamonds. This created the conditions for most effective use of the biological potential of fullerenes when interacting with microorganisms.

We also elaborated the necessary criteria for the usage of complexes. In additional tests, we studied the temperature and long-term stability of ternary and binary complexes (2.2). The optical density D(λ) was measured depending on the wavelength of light for the aqueous solutions of the complexes C60+PVP+AC960, C60+PVP, and C60+AC960 at the same concentrations as in the primary experiments (Figure 1), but under the conditions of heating from room temperature (TI = 23–24 °C) to 37 and 50 °C, followed by cooling to TF = 25 °C. In this temperature cycle, the samples showed only small changes in the D(λ) values. Appendix A represents the differences ΔDT(λ) = D(λ,T = 50 °C)-D(λ,TI) between the optical density magnitudes at 50 °C and the initial temperature for the three aqueous systems. Appendix A displays similar differences ΔDFI(λ) = D(λ,TF)—D(λ,TI) for samples at the final (TF = 25 °C) and initial temperatures (TI = 23–24 °C). For all samples, in a wide range of wavelengths (200–800 nm), the values of ΔDT(λ) ≤ 0.01 and ΔDFI(λ) ≤ 0.01 were very small, which confirms the stability of the binary and ternary complexes in water upon heating to 50 °C and subsequently being cooled down to room temperature. Then, we stored these samples for three months at room temperature (dark conditions) and repeated the optical tests. They showed the optical density of the solutions of the complexes, which changed by no more than 10%, demonstrating the satisfactory long-term stability of the complexes.

Along with this, we took into account that the biological efficiency of carbon nanoparticles may depend on their aggregation. Therefore, we measured the distributions of possible aggregates in size for C60+PVP+AC960, C60+PVP, and C60+AC960 complexes by nano-track analysis (NTA) (Figure 1C). We found that these complexes tend to form aggregates in the size ranges of 61 ÷ 102 nm, 33 ÷ 60 nm, and 69 ÷ 101 nm respectively. The C60+PVP complex had the smallest particle size.

### 3.2. Antimicrobial Activity Assay

In the first stage of the work, we studied the antimicrobial activity of the prepared complexes. The microbial contamination was found to be affected by the composition of the drug, its concentration, and exposure time (Table 1). At the same time, these factors may vary greatly for different types of microorganisms.

Similar consistent patterns have been noted previously. For example, the minimum inhibitory concentration of fullerene C60 functionalized with gentamicin was not the same for several bacterial species [51], and the effect of nanodiamonds on the viability of *S. aureus* depended on many factors, including the concentration and size of the nanoparticles, the suspension medium, and incubation time [24]. In [52], the authors demonstrated the different effectiveness of nanodiamonds against *S. aureus* and *E. coli*.

Along with this, the important characteristic of an antimicrobial agent is its ability to inhibit the formation of biofilms by microorganisms, which was analyzed in *S. aureus* and *E. coli* (Figure 2). All complexes prevented the formation of a biofilm, but their effectiveness varied greatly.

The activity of biofilm formation is represented in Figure 3 for *Staphylococcus aureus* and *Escherichia coli* (% relative to the control) as depending on the incubation time and sample concentration. It can be seen from the graph that all the samples, except for C60+PVP at the maximum used concentration, reduced biofilm formation in *Staphylococcus aureus* at a longer exposure time. The most effective there were the low concentrations of complexes containing nanodiamonds (C60+PVP+AC960 and C60+AC960). In the case of *Escherichia coli*, the complexes C60+PVP+AC960 (at a lower concentration) and C60+PVP (at a higher concentration) demonstrated the greatest effect. However, the effectiveness of C60+PVP did not increase with a longer incubation with it.

#### 3.2.1. Antimicrobial Activity of the C60+PVP+AC960 Complex

When analyzing the effect of the ternary complex on the growth of microbes, we observed the following regularities: C60+PVP+AC960 at a concentration of 350 µg/mL reduced the germination rate for all the tested microbes, while the efficacy against fungi of the genus Candida was low. The exposure time mattered only in the case of *B. subtillis*. The growth of these bacteria decreased by more than two times with an increase in exposure time from 45 to 90 min, and the greatest inhibitory effect was observed. With a twofold decrease in the concentration of the complex (up to 175 µg/mL), the effectiveness of the drug increased significantly. The seeding rate of Bacillus strains also depended on the time of exposure, but was much lower than at a higher concentration (at 45 min of exposure, it was 33 ± 11%, while at 90 min, it was even lower, at 6.7 ± 3.7%). The effect of the complex on *S. aureus* for both 45 and 90 min was accompanied by complete growth inhibition. At the same time, a low concentration of the complex had a sharply suppressive effect on *E. coli*, reducing the number of seeded colonies by three times after 45 min of exposure and by more than 10 times after 90 min. At a concentration of 175 µg/mL, the complex depressed biofilm formation well for *S. aureus* and *E. coli*, and it was also effective against Candida; their germination was low regardless of exposure time. The phenomenon of an increase in efficiency caused by a decrease in the concentration of nanoparticles obtained in this experiment was observed earlier. For example, it was shown in [53] that, upon the dilution of the polycarboxylic derivative of fullerene C60 in the aqueous medium, the relative number of aggregates decreased, while the number of single molecules increased, enhancing the activity of the compound per molecule.

#### 3.2.2. Antimicrobial Activity of the C60+PVP Complex

The effect of the binary complex consisting of C60 fullerene and PVP (which makes it possible to convert C60 into a soluble form) on *S. aureus* did not vary with the drug concentration, but inversely depended on the incubation time. With its doubling, the effectiveness of the impact also decreased. In contrast, for *B. subtillis*, 45 min of exposure was not enough to reduce microbial contamination. The increase in exposure time to 90 min was accompanied by the death of a half of the cells at a high concentration, and at lower one, the growth was 36 ± 7%. For Gram-negative *E. coli*, the highest concentration of the complex (300 µg/mL) and the maximum exposure time (90 min) were the most effective. In this case, the growth was completely suppressed. For fungi of the genus Candida, the lower concentration (145 µg/mL) with the increased exposure time proved to be the most effective. However, in other cases, the complex showed a good fungicidal activity. The C60+PVP complex had a minimal inhibitory effect on biofilm formation for *S. aureus*. However, the effect for the *E. coli* biofilm was effective, especially at high concentrations (50%).

#### 3.2.3. Antimicrobial Activity of the C60+AC960 Complex

When using the binary complex consisting of C60 and ND and existing as a stable suspension, the following consistent patterns were observed: For all the trial microorganisms, except of *S. aureus*, a higher concentration (200 µg/mL) was more effective than a lower one (100 µg/mL). At the same time, for *E. coli* and Bacillus, a longer exposure time provoked an increased effect, but for Candida, a reversed trend was observed. In the case of *S. aureus*, the exposure time was more important, as it was in the experiments with the C60+PVP complex. The same patterns we observed for biofilm formation.

As it follows from the results obtained, the response of the microorganisms to the complexes was not the same and depended on a number of factors. For Gram-positive bacteria, the ternary complex C60+PVP+AC960, containing both fullerene and nanodiamonds, showed higher efficiency (with increased incubation time for bacilli). The phenomenon of enhancement in the antimicrobial activity of nanoparticles against *S. aureus* by nanodiamonds was demonstrated previously [54,55]. However, with respect to Gram-negative *Escherichia coli*, the absence of a bactericidal effect of nanodiamonds has been reported [24,52]. In our study, nanodiamonds were effective, but the C60+PVP complex, which does not contain diamonds and is the most nontoxic for the cells as shown below, more actively reduced the growth, thereby preventing the formation of a biofilm of *E. coli*. The complex consisting of C60 and diamonds, without PVP stabilizing the colloid, had the least bactericidal properties, but limited biofilm formation for *S. aureus* and *E. coli*, probably due to the action on a biofilm.

It is known that one of the mechanisms of the death of microbes under the influence of carbon nanoparticles is a “direct contact”. Thus, the difference in the antimicrobial efficacy of carbon nanoparticles (CNPs) against Gram-negative and Gram-positive bacteria is associated with differences in the structure of their cell wall, while the size of NP aggregates can be of great importance [23]. In our case, the measurement of the average sizes of the aggregates showed no significant differences between the complexes. At the same time, we cannot say anything about the rate of aggregation under the conditions of biological media, since the measurements of such kind can only be conducted in water.

It was suggested in [56] that a lower efficiency of ND against *E. coli* can be associated not only with the composition of the cell wall, but also with the close values of the surface potential of the diamond and *E. coli*. The surface potential is an important parameter that can be regulated using such factors, such as surfactants, temperature, and pH. At the same time, it was shown in [57] that fullerenes demonstrated a strong binding ability to the outer membrane protein (OmpA) of Gram-negative *P. aeruginosa*. In our case, the contribution of the complexes to the antimicrobial activity against *E. coli* is also probably associated not only with the destruction of the cell wall of the microbe, but is mediated by the membranotropic properties of fullerenes and their high lipophilicity. This allows fullerenes to be integrated into membranes and penetrate into cell organelles, changing and disrupting their functions [58,59,60]. The effect of complexes on biofilm formation can be associated both directly with the bactericidal action and following the decrease in the number of microbes involved in the formation of the biofilm, and with the effect on the biofilm itself. For example, it is known that fullerenes prevent the aggregation of amyloid proteins, which are one of the components of the biofilm. Presently the effect of ND on this process is not fully explored.

Of particular interest is the antifungal activity of CNPs. Candida species cause various diseases of the oral cavity: caries, including in young children, and infections of the oropharyngeal mucosa [29]. C. glabrata is thought to have both innate and acquired resistance to antifungal drugs. The authors [28,61] discovered a concentration dependence of the effect of NDs on Candida albicans. In the analysis of our results, we have to state that all complexes had antimycotic properties to varying degrees, depending crucially on the preparation’s concentration range as well as the exposure time.

### 3.3. Analysis of the Toxicity of the Complexes in Human Cell Cultures

An important characteristic of any new drug is its low toxicity to normal human cells. Dermal fibroblasts are the main cellular component of the connective tissue layer of the skin. These cells provide skin homeostasis and its morphological and functional organization. They serve as the most adequate in vitro model for the analysis of promising drugs for wound and burn healing, as well as potential treatments for bacterial and fungal skin infections. Therefore, this test system was used in our experiments. To confirm the results, we duplicated the study on ECV cells, which have the properties of both endothelial and epithelial cells and are described as healthy transplantable cells.

According to the MTT test, the studied preparations differed in toxicity. Toxicity depended not only on the composition of the complex, but also on its concentration in the medium, on the time of exposure to the drug, and the nature of cells. During the 24 h incubation of cells in a medium that contained one of the complexes, C60+PVP+AC960, C60+PVP, or C60+AC960, the first complex, which contained all three components, showed the highest toxicity to fibroblasts (Figure 4A).

A concentration of 100 μg/mL in this case reduced the cell viability by approximately 20%. With a decrease in concentration, toxicity decreased: 10 μg/mL did not cause changes in cell viability. With increasing exposure time to 48 h, all the concentrations of this complex showed toxicity to fibroblasts (Figure 4B). However, the effect did not exceed 25% at the maximum concentration. It was of 15–20% at the concentrations of 50 µg/mL and 10 µg/mL, and became less than 10% for 1 µg/mL. However, the triple complex was only slightly toxic to ECV cells. Meanwhile, the C60+PVP and C60+AC960 complexes were less toxic compared to the ternary compound. With a 24 h incubation, only the maximum dose of 100 µg/mL (15–20%) showed toxicity, and exclusively for fibroblasts (Figure 4A,C). The 50 µg/mL dose also demonstrated toxicity after 48 h of drug exposure (Figure 4B,D).

With the antimicrobial activity of the complexes, higher concentrations of drugs (up to 350 μg/mL) were used with a shorter exposure time (45 and 90 min). To assess the response of DF2 and ECV cells under similar experimental conditions, an MTT test was made. The preparations C60+PVP and C60+PVP+AC960 at exposure times not exceeding 90 min were not toxic, even at the concentrations of 300 μg/mL. However, the C60+AC960, in this case, demonstrated some toxicity, but to fibroblasts only (Figure 5A–D).

The MTT test is a metabolic test and only indicates a decrease in cell viability, without providing an answer to the question of whether cells really die and how. We checked how the studied complexes affect apoptotic cell death. Apoptosis is one of the causes of cell death under the influence of damaging agents. To detect changes in membrane permeability associated with apoptosis, YO-PRO-1 dye, which does not accumulate in living cells, was used. In the early stages of apoptosis, ion channels are activated, which facilitates the transport of the dye into the cell. Propidium iodide passes through the damaged cell membrane; therefore, it is used to detect dead cells in the late stages of apoptosis and in the stage of necrosis. We analyzed the effect of 10 μg/mL and 100 μg/mL complexes at the time of their exposure to cells for 48 h (Figure 6). When exposed to drugs at a concentration of 10 μg/mL, we did not observe apoptotic cell death in the experimental samples, even in cases where the MTT test demonstrated toxicity (C60+PVP+AC960 complex). Upon incubation with 100 μg/mL of C60+PVP+AC960 and C60+AC960, cell death slightly increased and amounted to about 10%. The C60+PVP complex was the least toxic.

The addition of all complexes to the cell culture at the concentration of 1–100 μg/mL for 24–48 h did not show any morphological changes in the cells (Figure 7).

However, when the cells were cultivated with complexes containing ND, numerous inclusions were observed in the cytoplasm after 24 h, the number of which increased with the increasing concentration of the complex. When using C60+AC960, the particles were especially numerous both on the surface and inside the cells. The short-term incubation of the cells with the complexes for 60 and 90 min reduced the number of inclusions. It is believed that fullerenes quickly penetrate into eukaryotic cells by endocytosis, for example, clathrin-mediated endocytosis [62]. Earlier, we showed that 2 h after addition to the nutrient medium, hydroxylated derivatives of fullerenes, fullerenols, are detected in the cytoplasm of the cultured cells V 79 [63]. However, fullerene derivatives penetrate into cells at different rates. For example, the fullerene derivative F828 was detected in Hela cells only after 24 h [64], and a decrease in endocytosis was observed [65]. Presumably, in our case, the fullerene complexes on the diamond platform also penetrate the cells slowly; so, after 90 min, we see significantly fewer inclusions in the cytoplasm than after 24 h, regardless of the higher concentrations. Perhaps, this explains the non-toxicity of these complexes for cells under conditions similar to the assessment of antibacterial activity in the experiment. This fact is important, as it shows that, under conditions of short-term exposure, these complexes are effective against microbes, but less effective against human cells.

## 4. Conclusions

The data obtained during the experiments indicate that the studied complexes have low toxicity against the FD2 cell line, which is the most suitable for the study of anti-burn and wound-healing drugs. They do not cause apoptotic cell death; however, at high concentrations and with prolonged exposure, they cause metabolic disorders that are apparently not lethal.

It was established that all the studied complexes have a bactericidal effect, the severity of which depends on the type of microorganism tested, as well as on the chemical composition and the concentration of complexes, and exposure time. The effect on the microorganisms’ growth kinetics depended on the chemical nature of the preparation and the structural and metabolic features of the tested microbial cell.

According to the results of the tests, the complex C60+PVP+AC960 with a concentration of 175 µg/mL had the most stable, reliable microbicidal and inhibitory effect. Although, the amount of fullerene was minor in the dose of preparation (~1 wt. %); using the diamond platform, it was achieved a really high efficiency of fullerenes.

Such a significant effect of the preparation revealed, without optical activation, that fullerenes are the strongest photosensitizers. For new developed preparations, pronounced microbicidal photodynamic effects are expected, which could be studied further.

Thus, the complexes of fullerenes with PVP on the diamond platform obtained and successfully tested on bacteria will contribute to solving the problem of the effective suppression of pathogenic microorganisms and the disinfection of the environment, equipment, medical materials, and instruments, especially in medical institutions where microorganisms are resistant to serial antibiotics and diseases that occur there and are almost untreatable, leading to fatal outcomes in patients. Such complexes, which do not contain aggressive components, can be used to disinfect objects and materials that require a gentle and even delicate approach to their decontamination. The most significant are the prospects of the use of these drugs when excited by X-ray quanta for photodynamic therapy (XPDT) without restrictions on the penetrating power of activating radiation. In the next stages, we plan to develop prototypes of medical preparations based on diamond–fullerene complexes for a wide range of applications. This will require extensive studies of the mechanisms of antimicrobial action of new drugs in vivo, the study of various aspects of the use of drugs (safe dosages, methods of excretion from the body, and conditions for use in combination with other drugs). Thus, there is reason to consider the presented results as a real step towards the medical applications of fullerenes for the chemo-, radio-, and XPDT-therapy of socially significant diseases.

## Figures and Tables

**Figure 1 pharmaceutics-15-01984-f001:**
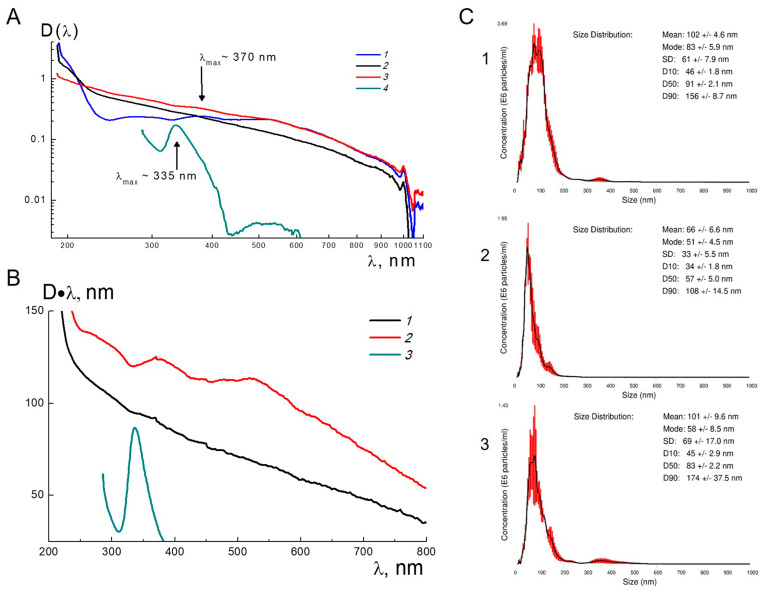
Characteristics of the complexes. (**A**) Optical density D(λ) vs. wavelength for the diluted aqueous solutions of the C60+PVP, C60+AC960, and C60+PVP+AC960 complexes (1–3; concentrations of 0.024, 0.019, and 0.033 mg/mL, respectively). For comparison, the spectrum of the solution of C60 in o-xylene (4) is shown. (**B**) Modified optical density D•λ vs. wavelength for the diluted aqueous solutions of C60+AC960 and C60+PVP+AC960 complexes (1–2; concentrations of 0.019 and 0.033 mg/mL, respectively). Similar data are plotted for C60 fullerene in o-xylene (3). (**C**) Size distribution and concentration profiles of C60-containing complexes by nano-track analysis (NTA). Red error bars indicate ±1 standard error of the mean. (1) C60+PVP+AC960; (2) C60+PVP; (3) C60+AC960.

**Figure 2 pharmaceutics-15-01984-f002:**
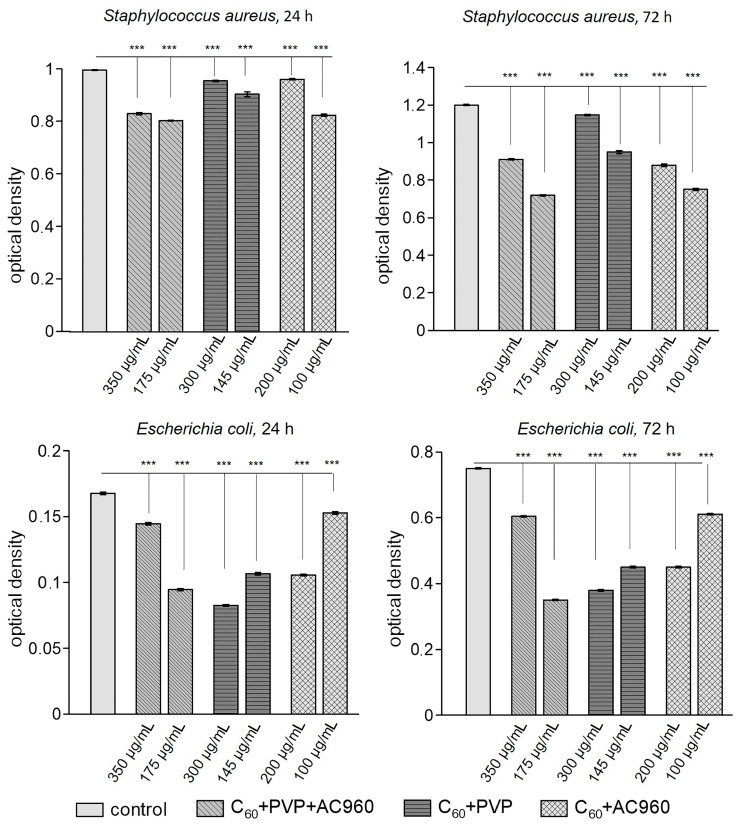
Effect of complexes containing fullerenes, nanodiamonds, and PVP on biofilm formation in *Staphylococcus aureus* and *Escherichia coli*. Different concentrations of the complexes and different incubation times with the preparation were used. *** *p* < 0.001; *n* ≥ 20–35 per point.

**Figure 3 pharmaceutics-15-01984-f003:**
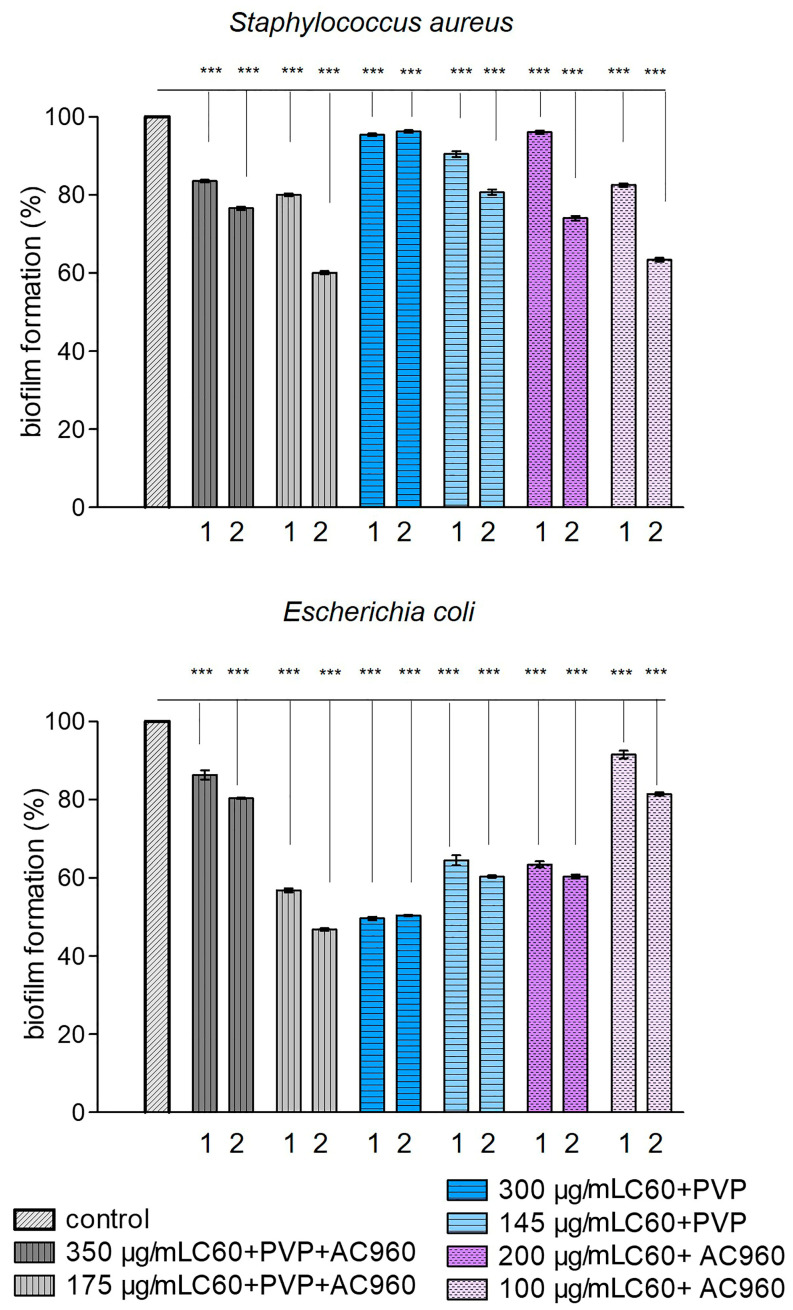
Activity of biofilm formation for *Staphylococcus aureus* and *Escherichia coli* (% of control) under the action of complexes containing fullerenes, nanodiamonds, and PVP. 1–24 h; 2–72 h. *** *p* < 0.001; *n* ≥ 20–35 per point.

**Figure 4 pharmaceutics-15-01984-f004:**
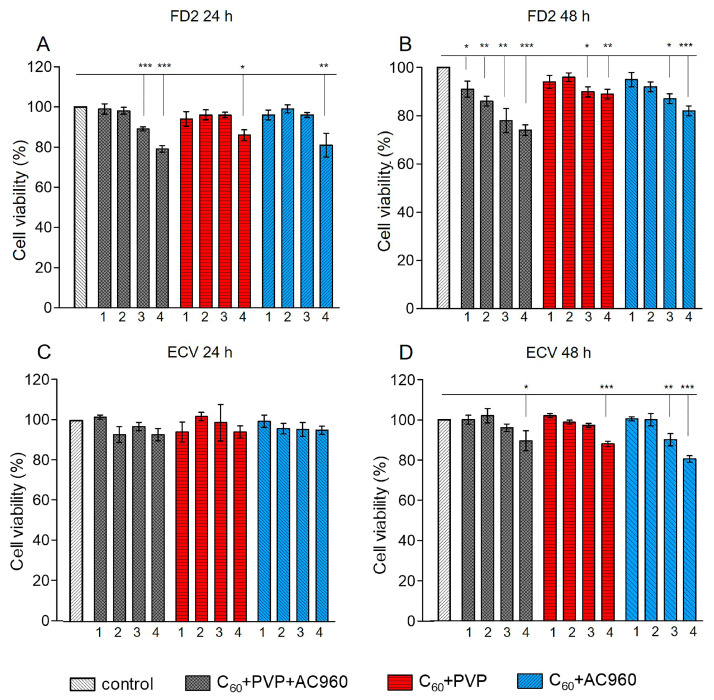
Analysis of the toxicity of complexes containing fullerenes, nanodiamonds, and PVP on the FD2 culture and ECV cells. Different concentrations of the complexes and different incubation times with the preparation were used (24 h and 48 h). (**A**) FD2 culture, incubation time with the preparation 24 h; (**B**) FD2 culture, incubation time with the preparation 48 h; (**C**) ECV cells, incubation time with the preparation 24 h; (**D**) ECV cells, incubation time with the preparation 48 h. Complex concentrations: 1–1 µg/mL; 2–10 µg/mL; 3–50 µg/mL; and 4–100 µg/mL. *** *p* < 0.001; ** *p* < 0.01; * *p* < 0.05; *n* ≥ 10 per point, in 3 separate experiments.

**Figure 5 pharmaceutics-15-01984-f005:**
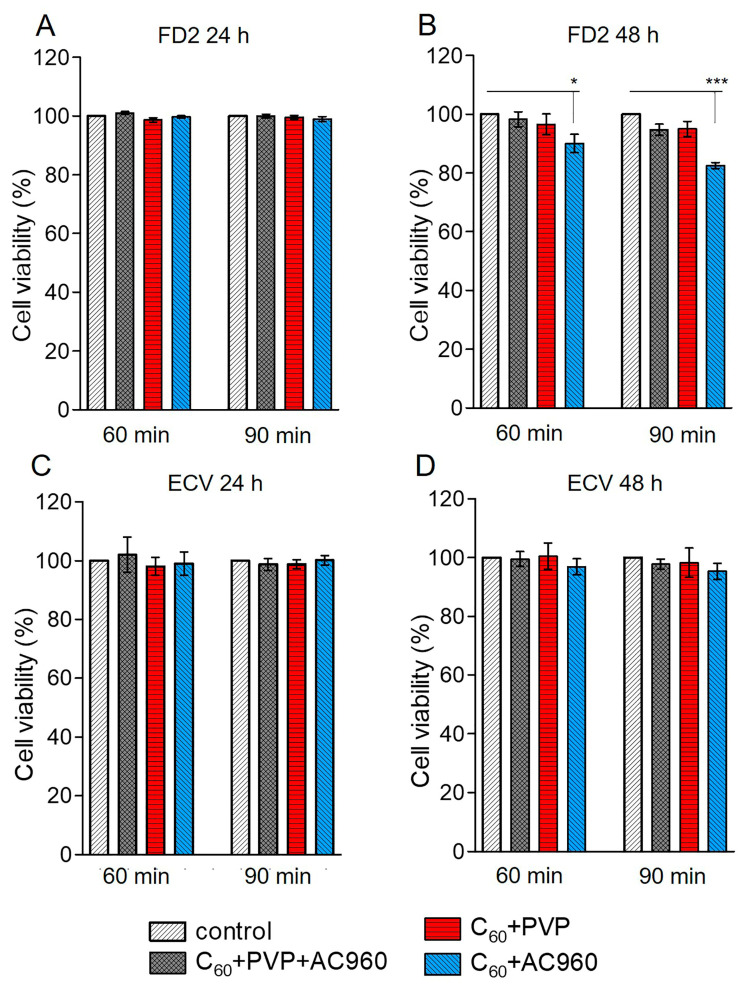
Toxicity analysis of complexes that contain fullerenes, nanodiamonds, and PVP at concentrations and exposure times comparable to experiments on microorganisms. (**A**,**C**): C60+PVP+AC960—175 μg/mL, C60+PVP—145 μg/mL, and C60+AC960—100 μg/mL. (**B**,**D**): C60+PVP+AC960—350 μg/mL, C60+PVP—300 μg/mL, and C60+AC960—300 μg/mL. *** *p* < 0.001; * *p* < 0.05; *n* ≥ 10 per point, in 3 separate experiments.

**Figure 6 pharmaceutics-15-01984-f006:**
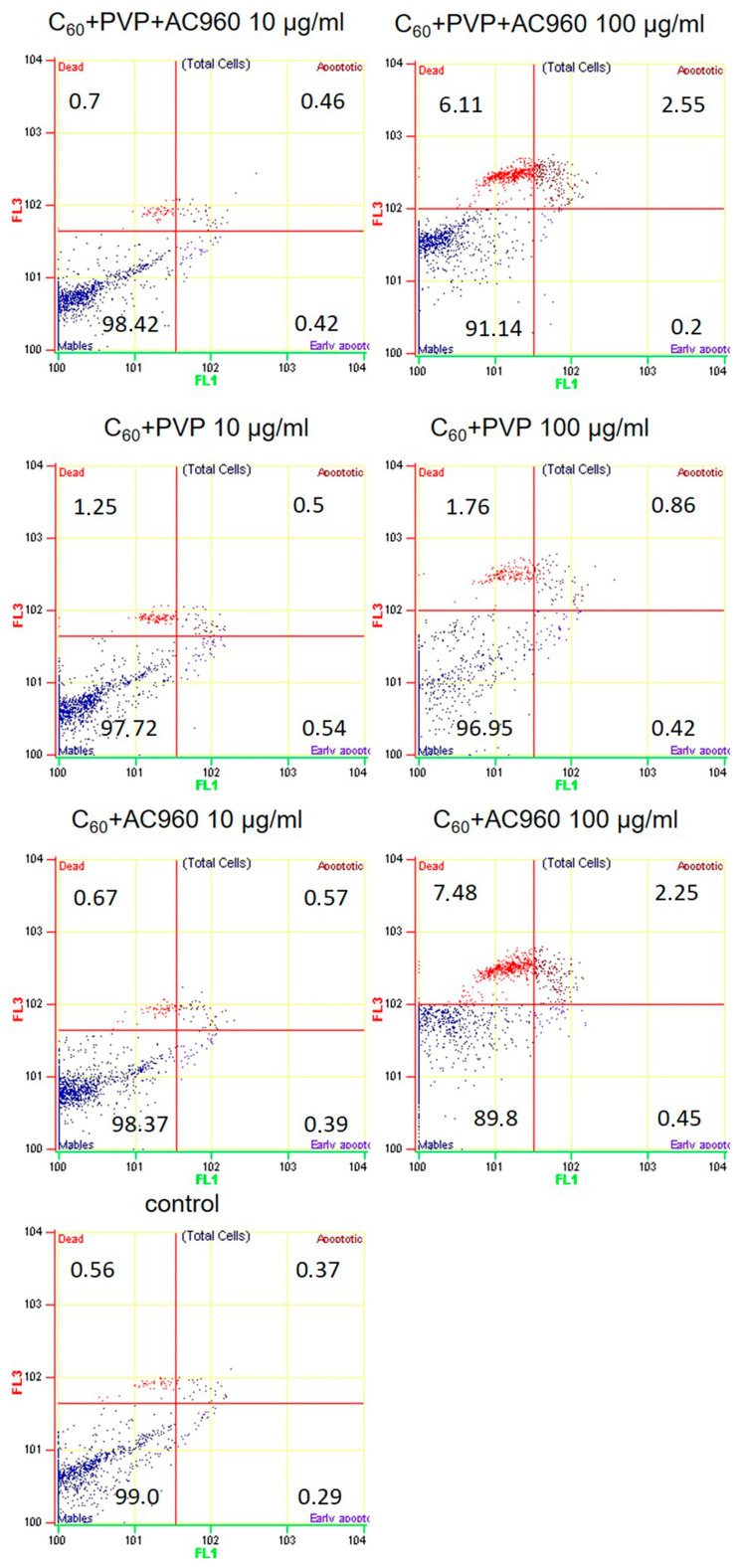
Effect of the complexes containing fullerenes, nanodiamonds, and PVP on the apoptotic death of human skin fibroblasts. The concentrations of drugs were 10 µg/mL and 100 µg/mL, and the exposure time was 48 h. *n* ≥ 3.

**Figure 7 pharmaceutics-15-01984-f007:**
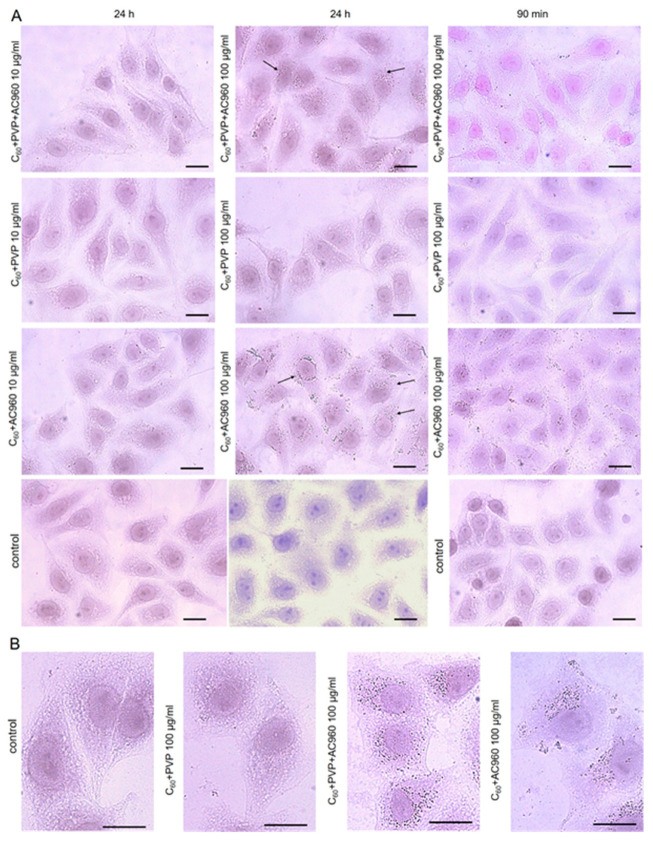
Effect of the complexes containing fullerenes, nanodiamonds, and PVP on the morphology of ECV cells. The concentrations of the drugs were 10 µg/mL and 100 µg/mL, and the exposure time was 24 h. Scale bar: 13.5 µm. (**A**) ×40 (air) objectives; (**B**) ×63 (oil) objectives. Arrows show inclusions in the cytoplasm.

**Table 1 pharmaceutics-15-01984-t001:** Characteristics of the seeding rates for various microorganisms under the influence of fullerene-containing preparations (%).

Type of Microorganism	Exposure Time	C60+PVP+AC960		C60+PVP		C60+AC960	
		350 µg/mL	175 µg/mL	250 µg/mL	125 µg/mL	200 µg/mL	100 µg/mL
*Staphylococcus aureus*	45 min	54.5 ± 3.1	0	36.3 ± 8.1	44.5 ± 4.4	33.6 ± 6.4	24.5 ± 3.7
	90 min	58.5 ± 5.3	0	64.2 ± 4.5	83.0 ± 3.2	69.8 ± 4.5	101.9 ± 4.9
*Bacillus subtillis*	45 min	66.7 ± 11.4	33.3 ± 11.1	105.6 ± 4.3	72.2 ± 10.5	27.8 ± 10.5	55.6 ± 11.7
	90 min	26.7 ± 6.6	6.7 ± 3.7	55.6 ± 7.4	35.6 ± 7.1	22.2 ± 6.1	40.0 ± 7.3
*Escherichia coli*	45 min	33.3 ± 5.3	30.8 ± 4.9	44.4 ± 3.7	15.0 ± 2.3	26.7 ± 3.7	58.3 ± 3.7
	90 min	41.7 ± 4.7	6.7 ± 3.7	0	18.3 ± 2.2	18.3 ± 3.7	58.3 ± 4.5
*Candida*	45 min	78.1 ± 7.3	16 ± 6.5	37.5 ± 8.5	43.7 ± 4.7	12.5 ± 5.8	31.3 ± 8.2
	90 min	95.2 ± 3.3	16.6 ± 5.7	40.5 ± 7.6	9.5 ± 4.5	28.6 ± 6.9	47.6 ± 7.7

Mean ± SEM.

## Data Availability

All data generated or analyzed during this study are included in the current manuscript.

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
