# Peer review of "Fullerenes on a Nanodiamond Platform Demonstrate Antibacterial Activity with Low Cytotoxicity"

_pharmaceutics, 2023, doi:10.3390/pharmaceutics15071984_

Round 1

Reviewer 1 Report

1. In the abstract there is a sentence “To overcome negative factors and enhance the action of fullerenes in an extended range of applications, for example, in antimicrobial photodynamic therapy, we have created new water-soluble complexes containing, in addition to C60 fullerene, purified detonation nanodiamonds (AC960) and/or polyvinylpyrrolidone (PVP).  ”.  Why authors didn’t test photodynamic antimicrobial activity  of complex ?

2.  In the abstract there is a sentence  “Complex C60+PVP+AC960 at a concentration of 175 mg/ml showed the most stable and pronounced inhibitory microbicidal/microbiostatic effect.” .  In the text later, authors mention value of 175 micrograms per milliliter(Fig 3 , page 10). Please write the correct value .

3. Introduction is not well organized and does not contain sufficient number of citations. State of the art of antimicrobial effects of fullerenes is not presented at all. There are numerous papers on antibacterial activity of fullerenes and fullerene derivatives which are not cited. Common mechanism is photodynamic action described in review papers such as “Biomedical potential of the reactive oxygen species generation and quenching by fullerenes (C60)” Biomaterials, 2008 Sep;29(26):3561-73. doi: 10.1016/j.biomaterials.2008.05.005..  Second segment should be presentation of the state of the art of interaction of bacteria and nanodiamonds. Third segment should be state of the art of C60/PVP  and nanodiamond/PVP complexes with adequate citations. I kindly ask authors to reorganize Introduction in this way.

4. Since C60 is extremely expensive antimicrobial material compared with other materials (chitosan or zinc oxide) please write few advantages of usage of C60 for antimicrobial purposes.

5. In the section 2.1. Preparation of the samples of carbon nanoparticles authors claim following “ The ternary colloid was dried to remove DMSO completely and further dissolved again in deionized water to produce the target product being aqueous colloid of C60+PVP+AC960 complex” . However C60 was dissolved in o xylene(line 75).  Did you remove o-xylene by drying too or not?

6. From personal experience, I know that it is very difficult to remove 100% o xylene and DMSO from produced complex by drying. Can you provide Raman analysis of C60+PVP+AC960 complex in water? O xylene have very strong Raman signal at 735 cm-1 while DMSO have peak at 2910 cm-1.

7. Section line 263-283 is speculation without scientific evidence. HRTEM or high resolution nanomechanical AFM is required to confirm distribution of C60 and AC960 in PVP matrix.

8. Nanodiamonds are highly luminescent in visible part of spectrum and often used for bioimaging. Can you provide photoluminescence spectra of AC960, PVP+AC960 and C60+PVP+AC960?

9. It is a bit strange that lower dosage of complex (175 micrograms per ml) is more effective than higher dosage (300 micrograms per ml) on biofilm formation (Fig2). Can you provide explanation? Did you keep biofilm in absolute dark or certain amount of light irradiated biofilms in 72 hours period? During antimicrobial tests did you keep samples in the absolute dark?

C60 nanoparticles are  extremely potent singlet oxygen generators even in ambient light conditions (“The mechanism of cell-damaging reactive oxygen generation by colloidal fullerenes” Biomaterials, 2007 Dec;28(36):5437-48. doi: 10.1016/j.biomaterials.2007.09.002)

Minor revision

Author Response

We sincerely appreciate the valuable comments and suggestions from reviewer. Please, find below our responses to the reviewers’ comments together with the revised versions of the manuscript and Supplementary Materials. The changes in the manuscript are marked with “Track Changes" function in Microsoft Word, so that changes are easily visible.

Reviewer 2 Report

In this work, the authors prepared a ternary complex C60+PVP+AC960. The results are interesting, and might be helpful for the development of antibacterial material. The complex seem to be rationally developed for antimicrobial properties with low toxicity to normal tissues. However, a number of issues need to be carefully addressed.

1.      It is advisable to clearly label the significance differences in Figure 2-5, indicating which groups they pertain to.

2.      The variations in the composition of the three samples you prepared are evident from the Materials and Methods section: Sample 1: С60 (11.1%) + PVP (44.4%) + АС960 (44.5%), Sample 2: С60 (20%) + PVP (80%), Sample 3: С60 (20%) + АС960 (80%). However, the subsequent experiments should use mass concentrations that encompass the entire mixture, meaning that the individual component proportions may differ. In light of this, can we establish comparability between the different groups?

3.      If the author intends to emphasize the individual contributions of each component in the ternary complex C60+PVP+AC960 to the antimicrobial effect, the control group should include AC960+PVP and PVP.

4.      Based on the author's results, it appears that the ternary complex C60+PVP+AC960 is not effective against all microorganisms. Therefore, from a conceptual standpoint, it would be more appropriate to specify the specific microorganism(s) it targets, rather than using the term "antibacterial activity" in a generalized manner.

5.      According to my knowledge, C60 is a black powder. However, based on the information provided, it seems that the author conducted the MTT experiment without performing a washing step after incubating the samples. This could lead to elevated absorbance values, potentially affecting the accuracy of the data. I kindly suggest the author carefully review the protocol to ensure that no steps have been omitted.

Author Response

(The authors gave the same response as above.)

Reviewer 3 Report

In the present manuscript, the authors have prepared the hydrophilic complex of C60, ND and PVP as antibacterial agent and have supported their findings with different cytotoxicity and antibacterial data making it a comprehensive and effective study. I have few suggestions:

On page 2, line 60, ND was written as HD by mistake and should be corrected.

On page 2, line 61, PVP was introduced without mentioning the benefits and advantages associated with it and what was the rationale behind using this polymer.

On page 3, why different concentrations of sample 1, 2 and 3 were used for antibacterial activity?

On page 3, line 123, what was the concentration and catalogue number of FGF used?

On page 7, line 256, C-60 needs to be corrected.

In Table1, the table font is not consistent and should be corrected.

On page 8, line 300, please check the relevance of the text "mean+SEM"

Authors are suggested to include the stability test of the complex.

In conclusion, the authors are suggested to include the future prospectives and potential of the complex along with any hurdles that might be caused in their clinical application.

Author Response

(The authors gave the same response as above.)

Round 2

Reviewer 1 Report

Dear Author,

in the list of references, text for reference 25 is missing.

Minor editing of English required

Author Response

We sincerely thank reviewer for valuable comments and suggestions.

  • in the list of references, text for reference 25 is missing.

Response: We have made changes in the list of references.

Reviewer 2 Report

Thank you for your response. I agree to publish it.

Author Response

We sincerely thank reviewer for valuable comments and suggestions